# Agility Training to Integratively Promote Neuromuscular, Cognitive, Cardiovascular and Psychosocial Function in Healthy Older Adults: A Study Protocol of a One-Year Randomized-Controlled Trial

**DOI:** 10.3390/ijerph17061853

**Published:** 2020-03-12

**Authors:** Mareike Morat, Oliver Faude, Henner Hanssen, Sebastian Ludyga, Jonas Zacher, Angi Eibl, Kirsten Albracht, Lars Donath

**Affiliations:** 1Department of Intervention Research in Exercise Training, Institute of Exercise Training and Sport Informatics, German Sport University Cologne, Am Sportpark Muengersdorf 6, 50933 Cologne, Germany; m.morat@dshs-koeln.de; 2Department of Sport, Exercise and Health, University of Basel, Birsstrasse 320 B, 4052 Basel, Switzerland; oliver.faude@unibas.ch (O.F.); henner.hanssen@unibas.ch (H.H.); sebastian.ludyga@unibas.ch (S.L.); 3Institute of Cardiology and Sports Medicine, German Sport University Cologne, Am Sportpark Muengersdorf 6, 50933 Cologne, Germany; j.zacher@dshs-koeln.de (J.Z.); a.eibl@dshs-koeln.de (A.E.); 4Institute of Biomechanics and Orthopaedics, German Sport University Cologne, Am Sportpark Muengersdorf 6, 50933 Cologne, Germany; albracht@dshs-koeln.de

**Keywords:** agility, prevention, health-related physical activity, healthy aging, community dwelling, multimodal exercise training, neuromuscular, cardiovascular, cognitive, psychosocial

## Abstract

Exercise training effectively mitigates aging-induced health and fitness impairments. Traditional training recommendations for the elderly focus separately on relevant physiological fitness domains, such as balance, flexibility, strength and endurance. Thus, a more holistic and functional training framework is needed. The proposed agility training concept integratively tackles spatial orientation, stop and go, balance and strength. The presented protocol aims at introducing a two-armed, one-year randomized controlled trial, evaluating the effects of this concept on neuromuscular, cardiovascular, cognitive and psychosocial health outcomes in healthy older adults. Eighty-five participants were enrolled in this ongoing trial. Seventy-nine participants completed baseline testing and were block-randomized to the agility training group or the inactive control group. All participants undergo pre- and post-testing with interim assessment after six months. The intervention group currently receives supervised, group-based agility training twice a week over one year, with progressively demanding perceptual, cognitive and physical exercises. Knee extension strength, reactive balance, dual task gait speed and the Agility Challenge for the Elderly (ACE) serve as primary endpoints and neuromuscular, cognitive, cardiovascular, and psychosocial meassures serve as surrogate secondary outcomes. Our protocol promotes a comprehensive exercise training concept for older adults, that might facilitate stakeholders in health and exercise to stimulate relevant health outcomes without relying on excessively time-consuming physical activity recommendations.

## 1. Introduction

The current report on the world’s aging population from the United Nations projected that, in 2050, nearly half of the world’s population will be living in countries with more than 20% of people aged 60 years and older [1]. Increasing life expectancy accelerates population aging [2], leading to considerable direct and indirect health care costs. Thus, healthy aging from individual and societal views has been entitled with “morbidity compression” [3] and includes multidisciplinary approaches with different physical, cognitive, social and psychological perspectives [4]. 

One strong precursor for successful aging is physical activity. Regular physical activity (approx. 150 min per week) can range from low exercise intensities to vigorous activities and can lead to notable risk reductions of non-communicable diseases and aging-associated diseases [5]. However, the adaptations to exercise training are usually very specific to the training modality and content [6]. In order to guarantee a broad range of training-induced adaptations in balance, strength, endurance, flexibility and cognition, exercise training recommendation should cover a variety of different multimodal tasks. Established exercise training guidelines for older adults are those of the World Health Organization [6] and of the American College of Sports Medicine [7]. Both associations independently recommend these physical training domains (i.e., aerobic exercise, resistance training, balance and flexibility training). Consequently, older adults are recommended to complete multiple training sessions of each domain. The total amount of these cumulative training sessions would result in seven weekly sessions. An expert panel additionally took into account the statements of the American Heart Association and published very similar recommendations [8]. This is a challenging and time-consuming physical activity guideline, which is particularly questionable, as “lack of time” is among the three most commonly mentioned burden for exercise participation in older adults [5]. Moreover, adherence rates for exercise training in older adults have been reported to decrease with every additional session per week [9]. 

Against this aforementioned background, Donath and colleagues [10] developed and proposed a multidimensional, functional and time-efficient agility training framework for older adults which has not been evaluated in a randomized controlled trial to date [10]. Their approach included changes of direction, start-stop movements, explosive and reactive strength and dynamic balance tasks. Agility training, therefore, comprises accelerations, decelerations, cutting maneuvers and different concentric and eccentric loads, combined with demanding spatial orientation tasks, strength and balance tasks in a very functional and integrative manner. Thereby, different levels of difficulty can be chosen for each exercise domain in terms of physical, perceptual, cardiovascular or cognitive demands. Overall, the agility training framework addresses all physical and selected cognitive health domains that are relevant throughout successful biological aging.

Agility is typically associated with sports games and usually comprises accelerations, decelerations, stop-and-go patterns, changes of direction (cutting maneuvers), and eccentric loads, combined with demanding spatial orientation tasks. When reviewing the scientific literature, very few studies have applied an agility-like training approach with older adults. The terminology “agility” is, more importantly, not present at all for this age group. However, relevant studies that performed group-based exercise training with older adults two to three times a week, comprising multimodal physical exercise training components and agility specific aspects like start-stop movements and cutting maneuvers, revealed meaningful improvements in strength [11,12,13], balance [14,15], cognitive [16] and endurance performance [12,14]. The greatest improvements were reported for functional mobility measures [12,13,17]. Thus, multimodal exercise training approaches have the potential to induce notable transfer effects to a variety of relevant physical exercise domains. However, those studies showed very heterogeneous training designs concerning the combination of training domains and the integration of agility-based training aspects. The first pilot study encompassing the agility approach for older adults, introduced by Donath et al. [10], includes an eight week-long agility program with high compliance (90 ± 8% of sessions) and notable improvements in endurance, balance and some strength measures [18]. This protocol of a randomized controlled trial was designed as a proof-of-concept training study that seems promising for a well-powered, long-term agility-based intervention study with different health-related surrogate measures.

In conclusion, it seems promising that agility training can serve as a feasible and time-efficient way to address a variety of health-related aspects relevant for the aging process. Numerous objective measures of neuromuscular, cardiovascular, cognitive and psychosocial origin can be challenged twice a week without excessive training volumes (which are not likely achievable by older adults). To our knowledge, no randomized controlled long-term intervention has applied the agility framework yet. Based on the above-mentioned rationales, we aim at introducing the protocol of an ongoing randomized controlled trial, investigating the effects of a one-year agility training intervention on neuromuscular performance, cardiovascular and cognitive functioning, as well as relevant psychosocial health outcomes in community-dwelling, healthy older adults. We specifically hypothesize that the agility training approach induces notable improvements in all physical performance measures (strength, balance, motor performance, aerobic capacity) after the one-year intervention period when compared to an inactive control group with at least moderate effect sizes that are comparable to other traditional training regimens. Secondly, we assume that those training effects will already be visible after six months of training with a further steady increase until the end of the one-year period. This study protocol was designed to contribute to a preventive multimodal training concept that can improve physical, cognitive and psychosocial health outcomes relevant to warrant healthy aging in an integrative and time-efficient way.

## 2. Materials and Methods 

### 2.1. Patient and Public Involvement

The study was designed without patient participation. After the end of the study, all participants will receive their individual results and the final publication. We intend to publish the results in a high-ranked, international journal. More importantly, we strive to produce a training app for coaches and therapists, which helps them to plan structured, progressive agility training sessions for elderly training groups.

### 2.2. Study Participants and Recruitment 

In the here-described ongoing one-year exercise training study, elderly participants, aged 60 years or older were recruited from the general population via a newspaper advertisement in Cologne (Germany). Older men and women were included if they were retired, community dwelling and independently living. They should not be engaged in more than two structured training sessions per week within three months before the start of the study. Their expected overall time of travel during the intervention period had to be less than two months. All participants had to accept randomization procedures to either the intervention or the control group. All participants needed to commute to the training site independently. Exclusion criteria were heavy smoking (“pack years” > 15), a body mass index (BMI) above 35 kg/m^2^ and a mini-mental state examination (MMSE) score below 26 [19]. Given contraindications for exercise training (any cardiovascular disease or depression without clearance from a medical doctor, chronic systemic inflammation or severe lung disease, insulin dependent diabetes, symptomatic cancer or acute cancer therapy, orthopedic diseases except those free of symptoms, more than mild, age-related osteoporotic changes) served as further exclusion criteria. Participants were informed about all inclusion and exclusion criteria prior to pre-testing. Additionally, they completed the “Physical Activity Readiness” questionnaire [20]. If they answered one or more questions with “Yes”, a medical consultation with their medical doctor was compulsory.

### 2.3. Experimental Design

The study is designed as a one-year two-armed, randomized, controlled intervention trial (Figure 1). All participants were requested to sign a written informed consent after being informed about all study procedures and after adequate time for consideration. Group assignment was conducted, applying the minimization method [21]. Accordingly, all participants were stratified to the intervention or the control group (intervention: control = 1:1). Sex, age, maximum knee extension strength, dual task gait speed and peak oxygen consumption (VO2peak) served as strata for balanced group allocation. Couples were stratified to the same group due to infrastructural, motivational and interference issues. The number of couples was evenly balanced in both groups. Group assignment was communicated to the participants after baseline assessment via telephone. Participants of the intervention group receive two weekly training sessions for one year. The control group does not receive any treatments. Only a brief written recommendation for physical activity is sent out. They will be given the opportunity to take part in a training program once a week after the end of the study for 6 months. Both groups are to keep a logbook on falls, diseases/injuries and exercise over the entire one-year study period.

Primary and secondary endpoints are collected at pre- and post-testing. An additional interim test with only a few measurements is provided after six months of training. At baseline (T1) and at the post-assessment (T3) after one year, all measurements are performed. This includes neuromuscular, cognitive, cardiovascular and psychosocial tests. At the interim assessment (T2), only selected neuromuscular and psychosocial measurements are assessed for reasons of time and reasonableness. 

T1 also includes a medical anamnesis and physical examination to rule out some severe exclusion criteria in addition to the medical clearance. At T1 and at T3, three lab visits on non-consecutive days, each lasting one to two hours, are required for each participant. On day one, cardiovascular, pulmonary and sonographic assessments are completed. On day two, neuromuscular, cognitive and psychosocial measurements are conducted. On the third day of assessment, the Agility Challenge for the Elderly (ACE) takes place. The interim assessments (T2) are completed within 45 min.

### 2.4. Endpoints and Assessment Procedures

Due to personal and financial resources, the blinding of assessors is not possible. This limitation provides a tolerable risk of bias. The same group of researchers assesses all measurements at baseline, interim and post-assessment. All measurements are conducted at the same laboratories with the same assessment tools at the German Sport University Cologne at the Department of Exercise Training and Sport Informatics (Intervention Research in Exercise Training) and the Institute of Cardiology and Sports Medicine. For each domain of assessment, an expert serves as a “gold-standard assessor” and instructs all other assessors about the measurement procedures. Accordingly, a standard operating procedure (SOP) can be assured throughout all measurements. T1, T2 and T3 are scheduled at the same time of the day if possible (at least morning and afternoon, respectively). Participants are asked to follow their usual daily routine on the testing day but without any strenuous exercise within 48h prior to testing. At T1, participants are tested with the MMSE for cognitive impairment and answer the “Freiburger Fragebogen zur körperlichen Aktivität” (FFB) [22] to assess baseline physical activity levels. Participant files are saved under non-personalized, pseudonymized identification numbers. A file for uncoding numbers is stored separately only accessible by the principal investigtor.

### 2.5. Primary Endpoints

According to the main components of the agility training approach, we selected neuromuscular measures of strength, balance and motor performance as primary outcomes. Maximum knee extension strength is measured under isometric conditions. Therefore, participants are seated in a leg extension machine (Edition-Line, gym80, Gelsenkirchen, Germany) that is equipped with a force transducer (mechaTronic, Hamm, Germany) with a hip angle of 90° and a knee angle of 120°. Participants are instructed to push as hard and as fast as possible against a pad that is placed above their ankle joints and to keep pushing for two to three seconds, following current recommendations on instructions for strength testing [23]. Maximum strength (Fmax) and the rate of force development (RFD) are computed by the measurement/analysis software IsoTest (version 2.0, meachTronic, Hamm, Germany). Thereby, Fmax serves as a primary outcome. Participants perform three trials, of which the best two are averaged for further data analysis. This testing procedure revealed high reliability for Fmax testing (r = 0.81) [24].

Reactive balance performance is tested barefoot on a two-dimensional platform (Posturomed, Haider Bioswing, Pullenreuth, Germany) as postural sway upon perturbation. The platform, which is mounted on eight springs, is initially fixed in a stable position. A lateral shift from the neutral position for 3.6 cm is initiated. After participants have taken a stable, two-legged, shoulder-wide stance with the hands on the hips (akimbo), slight knee flexion gazing a fixed spot on the wall (at 1.5m distance, 1.75m height), the fixation is suddenly and unexpectedly released after 2 to 5 s. Participants are initially instructed to reduce the induced pendular movement as fast as possible, and remain a stable position for 10 s. An acceleration sensor (MicroSwing® 6, Haider Bioswing, Pullenreuth, Germany) captures the amplitude of the pendular movement at 50 Hz. The total sway of the platform over 10 s is calculated and reported in mm (DigiMax Posturomed, Software Version 1.0, mechaTronic, Hamm, Germany). After one familiarization trial, participants perform two trials, of which the best is used for further analysis. Failed attempts are noted as seconds until failure. Overall reliability of unexpected perturbed balance measurements was reported as good (interclass correlation (ICC) = 0.75–0.89) [25].

For assessing dual task gait speed, participants are instructed to walk an eight meter-long distance at their habitual, comfortable walking speed. Additionally, they are given a number between 100 and 110 and have to count backwards in steps of three aloud. The gait speed of three trials is measured with photoelectric time gates (DLS/F03, Sportronic, Leutenbach-Nellmersbach, Germany) and counts and errors are manually noted. The mean value of all three trials is used for data analysis. A very high reliability of this test was assessed for this test for healthy older adults (ICC = 0.93) [26].

The newly developed ACE serves as an agility-specific, functional assessment. The setup is conducted according to Lichtenstein et al. [27]. Prior to testing, participants undergo a standardized warm-up (slow and brisk walking, knee lifting, walking backwards, hip rotation). Then, the ACE is demonstrated once. Participants are instructed to pass through the course as fast as possible without running. They complete the ACE four times and the first trial serves as familiarization trial. Total time and split times of the three components of the ACE are measured with photoelectric time gates (DLS/F03, Sportronic, Leutenbach-Nellmersbach, Germany), providing high reliability values for total and split times (ICC = 0.84–0.94) and a coefficient of variation of 4%–6.7% [27].

### 2.6. Secondary Endpoints

Further secondary endpoint assessment gives further insights in the effects of agility-based training and helps us to better understand the mechanisms of possible adaptations.

#### 2.6.1. Neuromuscular Assessments

Maximal leg curl strength assessment and maximal handgrip strength measurement are conducted in line with maximal leg extension strength testing. For the leg curl test, participants are positioned prone, with a hip angle of 170° and a knee angle of 120°. They are instructed to push against a pad that is mounted above their ankle joint. Reliability data of the described testing procedure of Fmax is considered high (r= 0.94) [24]. Handgrip strength is measured using a hand dynamometer (Digimax, Hamm, Germany). Participants take a stable upright stance, placing their left hand at their hip. They are then instructed to push the handles of the dynamometer together with their right hand. The stance height is adjusted to the participants’ height. Thus, the elbow is rectangularly and the forearm parallelly positioned to the ground. The grip width is adjusted to the size of the hand. A low standard error and high reliability are reported for the handgrip strength test for seniors (standard error of measurement (SEM) = 1.89) [28].

Jump height is assessed, employing the counter movement jump (CMJ). The CMJ serves as a measure of explosive strength of the lower extremities. Participants take a stable upright shoulder-wide stance on a force plate (FP4060-15 Bertec, Columbus, OH, USA) and place their hands on their hips. They are instructed to jump as high as possible, by reactively bending and stretching their legs. After two familiarization trials, three CMJ are recorded, whereas the averaged jump heights of the two highest trials is included into further data analysis. Jump height is computed by applying the impulse method using the software MR3 (version 3.10.64, Noraxon, Cologne, Germany). The test is classified as safe and can be used to test muscle function with a coefficient of variation of 6.6% for seniors [29].

Gait speed is assessed under single and triple task conditions in line with dual task gait speed. For the single task condition, participants walk in the above-described pattern without the additional cognitive task. In the triple task condition, in addition to counting backwards, they have to successfully carry a glass in one hand that is three quarters full of water.

Muscle architecture of the lower limb is assessed via muscle ultrasound by an experienced sonographer using a Vivid iq ultrasound machine, equipped with a 5 cm 3-9 MHz linear probe (GE Healthcare Systems, Boston, MA, USA, 2018) for the right M. gastrocnemius medialis (GM) and M. soleus (SOL) or with an Esaote ultrasound system, equipped with an 11 cm 3-10 MHz linear probe (Esaote Group, Florence, Italy, 2011) for the right M. vastus lateralis (VL), respectively.

The muscle architecture of the right GM is assessed on a participant in a prone position with the knee fully extended, foot placed against a wall with a rectangular position in the ankle joint. The knee joint space is palpated and marked with a skin marker. The Achilles tendon is tracked in a distal-to-proximal direction, from its insertion at the Calcaneus bone to the myotendinous transition of the M. soleus (this length is noted as length of the Achilles tendon) to the myotendinous transition of the M. gastrocnemius medialis. The distance of that last-mentioned landmark to the knee joint space is measured and the skin is marked at 50% of that distance. The medial and the lateral edges of the GM are identified by moving the probe medially and laterally along this plane. The distance is measured and the 50% distance is marked on the skin. This point represents the center of the muscle belly at 50% GM length. Three different longitudinal images of the GM muscle are taken. Additionally, three longitudinal images are taken at the myotendinous transition of the GM, also capturing the SOL beneath. 

The muscle architecture of the right VL is assessed on a participant in a supine position with the knee fully extended [30]. The foot is placed with the sole put against a wall, resulting in a rectangular position in the ankle joint. Landmarks of the VL are palpated (greater trochanter and lateral intercondylar notch) and marked with a dermographic pencil, the distance is considered as VL length. The distal 55% VL length is measured with a flexible tape measure. The operator then positions the probe perpendicularly to the main axis of the muscle at the marked 55% VL length. The medial and the lateral edges of the VL are identified by moving the probe medially and laterally along this plane. The full distance is measured and the 50% distance is marked on the skin. This point represents the center of the muscle belly at 55% VL length. Three different longitudinal images of the VL muscle are taken and saved for processing. Since the assessment of VL ultrasound is not yet well standardized in the literature, a second method is used to locate another region of the VL used by some other authors. Therefore, the greater trochanter and the tuberositas tibiae are used as landmarks, and the 50% distance is marked on the skin. Assessment of the medial and lateral edge and the subsequent steps is the same as mentioned above. Analysis of ultrasound images is performed with the free NIH Image-J software (Version 1.52a, U.S. National Institutes of Health, Bethesda, Maryland, MD, USA). Muscle thickness, fascicle length and pennation angle will be measured. 

Reactive balance performance, as described in the primary outcomes, is also assessed with the eyes closed. Static balance is measured recording the total sway of the center of pressure (COP) on a force plate (FP4060-15 Bertec, Columbus, OH, USA) at 1000 Hz in a one-legged stance and in a tandem stance position with eyes opened. In both conditions, participants need to hold their hands on their hips, gazing a fixed point on a nearby wall (1.5m distance, 1.75m height). They are instructed to stand in the given posture as still as possible for 10 seconds. The position of the COP is captured with Digital Acquir (version 4.12, Bertec, Columbus, OH, USA) and the path length is calculated with Matlab (version R2015b, Matlab, Natick, MA, USA). To avoid frequency noise, a sixth order Butterworth filter with a low-pass cut-off frequency of 10 Hz is applied. The calculation of the total path length is achieved in meters as follows [31]:$$\mathrm{COPpathlength}=\sum_{i}^{n}\sqrt{{\left(ax_{i}-ax_{i-1}\right)}^{2}+{\left(ay_{i}-ay_{i-1}\right)}^{2}}$$

After a few seconds of familiarization, three trials are recorded, of which the best is used for further analysis. Failed attempts are noted as seconds until failure.

#### 2.6.2. Cognitive Assessments

Modified versions of the N-Back and Erikson Flanker tasks are employed to assess the inhibitory control and working memory as two components of executive function. Both tasks are administered with E-Prime 3.0 (Psychological Software Tools, Pittsburg, PA, USA). Each task consists of a practice block and two test blocks. 

Visual stimuli in the N-back test are capital letters presented one at a time. For each letter, participants are instructed to indicate whether the actual letter matches the previous letter in the series by pressing a button with the left or right index finger. The adjusted hit rate and reaction time are calculated as outcomes. The results of a study published by Kearney-Ramos and colleagues [32] support the use of the n-back test as a viable and valid measure of working memory.

Visual stimuli in the Eriksen Flanker task are five arrows presented in a congruent or incongruent sequence. In congruent trials, the central target arrow faces in the opposite direction of the flanking arrows, whereas all arrows point in the same direction in incongruent trials. Participants are instructed to indicate the direction of the central arrow by pressing a button with the left or right index finger. Reaction time (on response-correct trials) and accuracy are calculated as outcomes [33].

The mini-mental state examination (MMSE) is employed at baseline to rule out cognitive impairment. The MMSE assesses orientation, memory, attention, calculating, writing and speech and showed good reliability (r = 0.89) [34].

#### 2.6.3. Cardiovascular Assessments

All cardiovascular assessments, except the retinal vessel analysis, are performed by a medical doctor or by qualified personnel under a doctor’s supervision.

To evaluate the aerobic capacity of the participants, a spiroergometric test on a cycle ergometer (Ergoline, Bitz, Germany) with maximal oxygen consumption (VO2max) (Cortex Metamax, Leipzig, Germany) and electrocardiogram (ECG) (Amedtec, Aue, Germany) measurements are conducted. The test is started at 25 W with an increment of 25 W every three minutes until subjective exhaustion is reported. Thereby, common exhaustion criteria were applied [35]. At the end of each level, heart rate and blood pressure are measured and a capillary blood sample is drawn from the ear lobe to measure and evaluate lactate concentrations. Participants are asked to state their perceived exertion on the BORG scale after every level [36]. For spiroergometric assessment, the peak and relative peak oxygen consumption related to body weight (VO2peak) and the maximum power (Wpeak) are assessed.

Pulse wave velocity (cfPWV) is assessed as a measure of arterial stiffness using an oscillometric device with integrated ARCSolver® software (Mobil-O-Graph® Monitor, I.E.M. GmbH, Stolberg, Germany). The measurements of arterial stiffness and central hemodynamics using the oscillometric method stand in good agreement with the conventional tonometric method [37]. All measurements are performed in a fasting state. Patients are required to refrain from physical exercise, eating as well as drinking alcohol or caffeine 12 h prior to measurements. The blood pressure cuff is placed on the left upper arm while the patient is lying in a resting supine position. After 10 minutes of rest, three measurements are performed with two-minute intervals in between. From the measurements, central blood pressure, the augmentation index at heart rate 75/min (AIx@75) as well as cfPWV are extracted. cfPWV is automatically calculated from the pulse wave data and reflects the assessment of central, aortic arterial stiffness. After data readout, every measurement is reviewed for erroneous values. The mean and standard deviation of three valid measurements are calculated. 

Furthermore, heart rates, heart rhythm and heart rate variability are documented using a 24 h electrocardiogram (ECG) (Langzeit-EKG, Amedtec, Aue, Germany). Blood pressure is documented over 24 hours every 30 minutes during the day and every 60 minutes during the night using a standard cuff (Mobil-O-Graph®, IEM, Stolberg, Germany).

A non-invasive retinal vessel analysis is conducted to measure retinal vessel diameters as microvascular biomarkers of cardiovascular risk using a fundus camera and a semi-automated software system (SCA-T, Imedos Systems UG, Jena, Germany). Three images of the right eye are taken at a 45° angle with the optic nerve at center. Arterioles and venules at a distance of 0.5–1 times the diameter of the optic nerve are semi-automatically analyzed (Vellelmap 2, Visualis, Imedos Systems UG, Jena, Germany). The high reliability of this method has been shown [38,39]. 

Echocardiography is performed by an experienced echocardiographer using the Vivid iq with a M5Sc 1.5-4.6 transducer (GE Healthcare Systems, Boston, MA, USA, 2018). The participant lies in a semi-supine resting position. Images are taken based on the recommendations of the American Society of Echocardiography (ASE) while the participant holds his breath [40]. Conventional echocardiography includes standard measurements of cardiac dimensions, contractility and diastolic function. Speckle tracking images for calculation of myocardial strain are recorded in apical three-chamber, two-chamber and four-chamber views for longitudinal values and in the parasternal short-axis at the level of papillary muscles for circumferential and radial values. Analysis of all images is conducted offline using EchoPac-Software (Version 202, GE Healthcare, Boston, MA, USA, 2018).

#### 2.6.4. Psychosocial Assessments

Participants complete various questionnaires in written form. 

Depressive symptoms are assessed using the short “Centre for Epidemiological Studies Depression Scale” (CES-D) with 15 items, containing depression related cognitive, emotional, motivational, somatic and behavioral aspects [41]. The validity of the CES-D has been shown [42].

The German version of the “Perceived Stress Scale” (PSS) with ten items is used to inquire the perceived stress of the past months [43]. Its validity and reliability have been shown [44] and the scale is commonly used with older adults [45].

To quantify sleep complaints, participants answer the “Insomnia Severity Index” (ISI) [46]. A positive evaluation of its reliability (Cronbach α = 0.9) is published [47] and it is often used with older adults [48].

A short German version of the “World Health Organizations Quality of Life Questionnaire” (WHOQOL BREF) is also applied in order to assess global measures of quality of life [49]. Internal consistency measures were high (Cronbach’s alpha = 0.9) [50].

The fear of falling is recorded using the “German Version of the Falls Efficacy Scale—International Version” (FES-I) [51], which is a valid and reliable scale [52].

### 2.7. Interim Assessment

The following measurements are only applied at the interim assessment: Maximal strength assessment of leg extension, leg curl and handgrip; gait speed (in single-, dual-task- and triple-task conditions); balance (static and perturbed); psychosocial questionnaires (CES-D, PSS, ISI, WHOQOL).

### 2.8. Intervention

The first training session takes place immediately after group allocation. Participants train in three separate groups with a maximum of 13 participants in each group. Two trained study assistants supervise all sessions. Each of the training groups train twice a week for one year with a two-week Christmas break. Training sessions are held at gyms, located at the campus of the German Sport University Cologne. The gyms are equipped with regular basic equipment (e.g., mats, cones, ropes, hoops). The sessions last for 60 min in total and consist of a diverse 10 min agility-specific warm-up, followed by a 45 min agility training and a 5 min cool-down. Training is accompanied by music in line with the preferences of the participants. The warm-up either consists of a series of instructed exercises, dance-like routines or exercises incorporating balls. The agility training part is divided into three to four stations that the participants attend in pairs or in small groups. The load duration is one minute, followed by one minute of rest with four to five repetitions at each station. In the last third of the one-year intervention period, more complex parkour settings are installed. The individual load duration exceeds one minute and the time for rest is longer. As briefly described in the introduction, agility training includes four components: start-stop, change-of-direction, balance and strength. In order to progressively design training sessions, the one-year intervention period is divided into three thirds. During the first third, two of the four agility components are part of each training session, alternating every second session. In the second third, three of four components are combined in every session and then, only in the last third, all agility components are simultaneously integrated in each session. The complexity of the training sessions is progressively increased by changing the physical, perceptual and/or cognitive loads of the exercises. The training design and progression are displayed in Figure 2. Participants are always pointed to potential hazards during the exercises (e.g., stumbling over mats, slipping on a cone). They are taught to give each other assistance, e.g., during balancing exercises. Any adjustment of the original training protocol as well as adherence for every session is documented in detail by the coach. For each session, participants are asked to state their overall rate of perceived exertion (BORG scale) immediately after the end of the session. Additionally, in each session, three to four randomly chosen participants wear heart rate sensors (polar, H7, Buettelborn, Germany) to exemplarily capture heart rates during the training. A participant does not continue the allocated intervention if he or she requests it or suffers pain or an injury. Adverse events are recorded and potential sources will be discussed and eliminated if necessary.

### 2.9. Sample Size Estimation and Statistical Procedures

A sample size estimation was conducted with G*Power (Version 3.1, Heinrich Heine University, Duesseldorf, Germany). 68 participants (34 per group) are needed for statistical analysis, assuming moderate effects (f = 0.2) for training-induced changes in the primary outcomes (statistical power 90%, two-sided significance level α=.05). Expecting a realistic dropout rate of about 25%, a total of 85 participants were initially recruited.

Data entry will be double-checked and data ranges will be controlled for plausibility. Two researchers (MM, LD) will do data management and analysis. Independently conducted two-factorial repeated measured analyses of co-variance (rANOVA), with the respective endpoint as a baseline measure serving as the covariate [53], will be computed as primary analysis to assess the time course of adaptations between groups (factor 1: control group vs. intervention group; factor 2: time point of measurement; baseline, interim, post). The rANOVA interaction term between both factors is the main term of interest. Together with 90% confidence intervals, change scores from T1 to T3 and from T1 to T2 and T2 to T3 are calculated to arrive at an estimate of the size of the observed effects. The intention-to-treat analysis will be computed. The intention-to-treat analysis will be compared to as-treated analysis. Potential differences will be discussed.

### 2.10. Ethics and Dissemination

All subjects gave their informed consent for inclusion before they participated in the study. The study was conducted in accordance with the declaration of Helsinki and was approved by the local ethics committee (Cologne, Germany; 131/2018). It has been registered in drks.de (DRKS00017469 registered June 2019; drks.de identifier DRKS00017469 (https://www.drks.de/drks_web/navigate.do?navigationId=trial.HTML&TRIAL_ID=DRKS00017469)). All results will be published in international peer-reviewed journals. We intend to produce a training app for coaches and therapists, which helps them to plan structured, progressive agility training sessions for elderly training groups. Authors have no competing interest to declare.

## 3. Discussion

### 3.1. Expected Key Results

The agility training framework [10] seems to be a promising and feasible framework to time-efficiently address multiple relevant health aspects in the aging process. However, the specific approach, as introduced by Donath et al. [10], has only been examined in one pilot study so far [18]. As Donath et al. [10] already stated, the best practice recommendations of exercise training for older adults do not cope with the interplay of neuromuscular performance, cognitive function and cardiovascular performance. The agility framework does, by promoting an appealing, time-efficient training for older adults that is adaptable to real life settings. It is, therefore, more functional than common training settings and more specific to situations where, e.g., balance is threatened in real life. Lichtenstein et al. [27] conducted a pilot study with first promising effects of agility training as a basis for a long-term intervention study [18]. They conclude that agility training might lead to favorable adaptations in muscle power, endurance, balance and strength. They see agility training as a time-efficient alternative for exercise training for older adults, as all relevant aspects of human performance in aging are simultaneously trained. However, the authors also state that training load needs to be captured in terms of the rate of perceived exertion (RPE) or heart rate in future studies. They emphasize the need for a long-term evaluation of agility training on objective measures of neuromuscular, cardiovascular, cognitive and psychosocial origin in older adults. We think that by applying a one-year agility training intervention, healthy older adults can experience relevant improvements in neuromuscular performance, cardiovascular and cognitive functioning, as well as in psychosocial health outcomes, compared to an inactive control group. These improvements will be visible in the interim assessments and will increase until the end of the one-year training period. Positive changes in strength, balance, motor performance, cardiovascular capacity and cognitive function are highly relevant for activities of daily living and will come along with enhanced psychosocial health.

### 3.2. Benefits and Risks

Participants who take part in the agility training program have potential benefits from regularly participating in group-based exercise training, aiming at improving their overall health. No adverse events are expected during the measurements or the intervention based on the experiences of the pilot trial [18]. Two coaches for a maximum of 13 participants closely supervise all training sessions. Participants are always pointed to potential hazards and learn to assist each other, e.g., during balance exercises. Coaches ensure the proper execution of all exercises. Due to the medical screening and carefully selected inclusion and exclusion criteria, all participants are expected to fulfil the prerequisites for training participation. Individual adjustment of exercises according to the participants’ abilities is always possible with the help of the coaches. All measurements are performed by well-trained personnel and if necessary by a medical doctor, so that no risk is seen in any assessment procedure. Overall, expected benefits are exceeding the potentially occurring risks. Proper safety precautions additionally minimize the risk of uncomfortable and adverse events.

### 3.3. Potential Limitations and Risk for Bias

Participants were recruited from the general population via newspaper advertisement, which could lead to a selective recruitment of those older adults, that are intrinsically motivated to exercise and wo are already physically active. Group allocation was done according to the minimization method, which is comparable to a randomization method in smaller samples [21]. It allows us to balance important demographic measures and primary and secondary outcomes with possibly small group differences at baseline. Sex, age, maximum knee extension strength, dual task gait speed and VO2peak serve as strata for minimization, notably attenuating the risk of bias. Only participants who stated that they were willing to be randomized in one of the two groups could take part in the study to minimize dropout in the control group. Group preferences were strictly not allowed. Couples were randomized to the same groups. As is usual in exercise intervention studies, the blinding of participants is not possible. Due to personal and financial resources, the blinding of assessors and coaches is not possible either, which yields an acceptable risk for bias. Measurements are standardized and performed in the same room by the same assessor at T1, T2 and T3. Measurements are scheduled at the same time of the day for T1, T2 and T3 if possible. For each measurement domain, one gold-standard setter teaches all other assessors. The control group does not receive any treatments, but for ethical reasons they receive written recommendations for physical activity. Their potential autonomous exercise over the one-year period could influence the results of the study and is, therefore, captured via a logbook.

## 4. Conclusions and Perspectives

The results of this study might establish the agility training approach for older adults and propose a well-studied concept for training. This would provide new perspectives for older adults to time-efficiently train various exercise-based health issues adequately. The ultimate aim of this study is to publish the results in a high-ranked peer-reviewed scientific journal to make it accessible for the scientific and general community. More importantly, we aim to distribute the concept to coaches of elderly training groups in the form of an app to reach out to the end-user.

## Figures and Tables

**Figure 1 ijerph-17-01853-f001:**
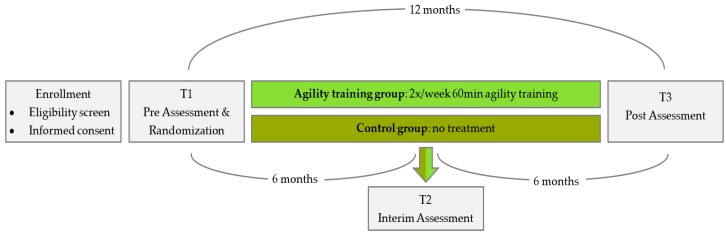
Study design.

**Figure 2 ijerph-17-01853-f002:**
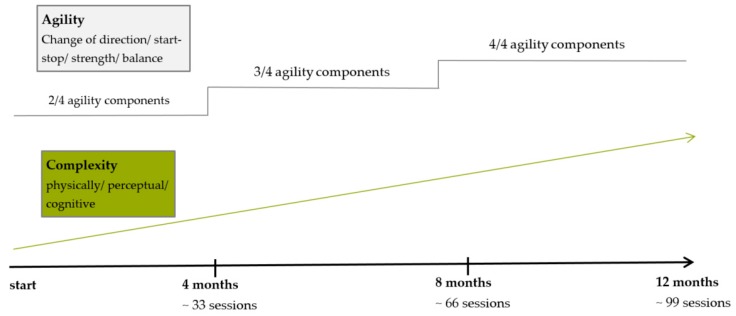
Training design and progression.

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
