# Peer review of "Agility Training to Integratively Promote Neuromuscular, Cognitive, Cardiovascular and Psychosocial Function in Healthy Older Adults: A Study Protocol of a One-Year Randomized-Controlled Trial"

_ijerph, 2020, doi:10.3390/ijerph17061853_

Round 1

Reviewer 1 Report

The manuscript is well-written and structured. The theoretical basis is satisfactory and justifies the study. The references are current and pertinent to the theme of study.

Specifically on page 3, lines 93-107, I suggest writing more clearly that the purpose of the manuscript is only to describe the design of a study protocol of a one-year randomized-controlled trial involving the agility-training. As it stands, there is an expectation of presenting your results.

The methods described are innovative and consistent with the study proposal. However, I did not understand the use of WHOQOL_Bref to analyze the quality of life of the elderly. Why the WHOQOL_ Old was not used?

The discussion is well written and meets expectations; however, I did not identify the prediction of any limitations on future study findings.

Author Response

Dear Editor, dear Reviewers,

Thank you very much for your positive feedback and your constructive comments. We appreciate your time and effort to improve the manuscript. In the following, we address all issues raised using a point-to-point reply. The point-to-point answers are in Italics, referring to the lines, where the changes have been made. We now feel that the manuscript notably improved and hope that the paper can be accepted for publication.

I hereby reconfirm the author list and the corresponding affiliations.

Thank you very much

Mareike Morat on behalf of all co-authors

Reviewer 1

The manuscript is well-written and structured. The theoretical basis is satisfactory and justifies the study. The references are current and pertinent to the theme of study.

Specifically on page 3, lines 93-107, I suggest writing more clearly that the purpose of the manuscript is only to describe the design of a study protocol of a one-year randomized-controlled trial involving the agility-training. As it stands, there is an expectation of presenting your results.

Reply: Thanks a lot for raising this important point. In line 101-102, we specified that “… we aime at introducing the protocol of an ongoing randomized controlled trial…”. In line 110 we also changed “Our study” to “This study protocol” to make it clear.

The methods described are innovative and consistent with the study proposal. However, I did not understand the use of WHOQOL_Bref to analyze the quality of life of the elderly. Why the WHOQOL_ Old was not used?

Reply: Thank you for bringing up this question. Due to our experience, the WHOQOL_Bref is an establiseh questionnaire to assess QOL, which has already been successfully employed in a couple of studies in our research group with older adults. Both questionnairs include similar domains. Moreover, the WHOQOL_Bref was reported to provide higher internal consistency scores compared to the WHOQOL_Old, depicted by Cronbach’s alpha. (Lucas-Carrasco et al., 2011). Thus, we decided to use the WHOQOL_Bref. In line with your remarks, we added Crombach’s alpha values to the manuscript (line 369-370: “Internal consistency measures were high (Cronbach’s alpha = 0.9) [50]”. We now feel that this part notably improved.

The discussion is well written and meets expectations; however, I did not identify the prediction of any limitations on future study findings.

Reply: Thank you for this comment. We changed the heading in line 466 from “Potential for bias” to “Potential limitations and risk for bias”. One potential limitation that we already address in line 476-478 is that “… blinding of participants is not possible. Due to personal and financial resources, blinding of assessors and coaches is not possible either, which yields an acceptable risk for bias”. In line 467-469 we added “Participants were recruited from the general population via newspaper advertisement, which could lead to a selective recruitment of those older adults, that are intrinsically motivated to exercise and wo are already physically active.” Furthermore we added “The control group does not receive any treatments, but for ethical reasons they receive written recommendation for physical activity. Their potential autonomous exercise over the one year period could influence the results of the study and is therefore captured via a logbook.” in line 481-483.

Reviewer 2 Report

Interesting study. The abstract suggests data may have already been collected but article is more about promoting the proposed protocol. For example the last sentence is misleading: "Our results promote a comprehensive exercise training concept for older adults, facilitating stakeholders in health and exercise to promote relevant health outcomes without relying on excessively time consuming physical activity recommendations". Apart from more clearly stating that article is meant to promote a concept and a protocol vs a presentation of completed study, the article is well done. The rationale is well presented, methods clearly described and the battery of testing is comprehensive. The writing style, for the most part, is polished. Potential limitations are acknowledged. I'd look forward to the completed study.

Author Response

Dear Editor, dear Reviewers,

Thank you very much for your positive feedback and your constructive comments. We appreciate your time and effort to improve the manuscript. In the following, we address all issues raised using a point-to-point reply. The point-to-point answers are in Italics, referring to the lines, where the changes have been made. We now feel that the manuscript notably improved and hope that the paper can be accepted for publication.

I hereby reconfirm the author list and the corresponding affiliations.

Thank you very much

Mareike Morat on behalf of all co-authors

Reviewer 2

Interesting study. The abstract suggests data may have already been collected but article is more about promoting the proposed protocol. For example the last sentence is misleading: "Our results promote a comprehensive exercise training concept for older adults, facilitating stakeholders in health and exercise to promote relevant health outcomes without relying on excessively time consuming physical activity recommendations".

Reply: Thank you for addressing this point. We made the following adjustments ins the abstract: line 24-25 “The presented protocol aims at introducing an two-armed, one-year randomized controlled trial, evaluating the effects of this concept…”; line 30 “The intervention group currently receives supervised, group-based agility-training…”; line 34-36 “Our protocol promotes a comprehensive exercise training concept for older adults, that might facilitate stakeholders in health and exercise stimulate relevant health outcomes without relying on excessively time-consuming physical activity recommendations.”

In the introduction, (line 101-102), we specified that “… we aime at introducing the protocol of an ongoing randomized controlled trial…”. In line 110 we also changed “Our study” to “This study protocol” to make it clear.

Apart from more clearly stating that article is meant to promote a concept and a protocol vs a presentation of completed study, the article is well done. The rationale is well presented, methods clearly described and the battery of testing is comprehensive. The writing style, for the most part, is polished. Potential limitations are acknowledged. I'd look forward to the completed study.

Reply: Thanks a lot for this positive feedback on our work.

Reviewer 3 Report

The authors present an interesting idea from the point of view of training with older. Anyway, I have some points that should be revised before publication.

The link for the protocol access is wrong. Please, revise it in order to allow the reviewers to check the protocol registration. Please, take on account that the protocol should be written in future tense, as it is not implemented yet. When comparing an experimental group with a control group, commonly the EG is better regarding almost every parameter measured than the CG. In this way I recommend to reconsider the idea of including another multicomponent-based training experimental group in the comparison (two experimental groups and a control group). In this way you could probe if your agility protocol is better than the traditional way of training nowadays, having a control group as a base reference. The difference of your multicomponent training and the other multicomponent training should be well defined. If this is a protocol for RCT, the primary outcome should be only one variable. Please, revise. I recommend to follow the SPIRIT checklist for protocols and to use the TIDIER checklist to describe the intervention.

Author Response

Dear Editor, dear Reviewers,

Thank you very much for your positive feedback and your constructive comments. We appreciate your time and effort to improve the manuscript. In the following, we address all issues raised using a point-to-point reply. The point-to-point answers are in Italics, referring to the lines, where the changes have been made. We now feel that the manuscript notably improved and hope that the paper can be accepted for publication.

I hereby reconfirm the author list and the corresponding affiliations.

Thank you very much

Mareike Morat on behalf of all co-authors

Reviewer 3

The authors present an interesting idea from the point of view of training with older. Anyway, I have some points that should be revised before publication.

The link for the protocol access is wrong. Please, revise it in order to allow the reviewers to check the protocol registration.

Reply: Thank you for checking. I reentered the link and it correctly links me to the respective protocol in the german clinical tral register. So unfortunatelly, I can not see what went wrong, when you tried to access it.

Please, take on account that the protocol should be written in future tense, as it is not implemented yet.

Reply: Thank you for addressing this point. As I stress in the abstract and in the introduction, it is a protocol of an ongoint trial, meaning that pre assessment and group allocation have already been done (line 27 “Eighty-five participants were enrolled in this ongoing trial”, line 101-102 “…introducing the protocol of an ongoing randomized controlled trial“. Participants are currently in the intervention phase (line 30 “The intervention group currently receives supervised, group-based agility-training”). Guided by another protocol of an ongoing study that was recently publised in this journal (Fischer et al. 2019), we therefore chose to consequently address recruitment and group allocation in past tense and measurements and intervention in present tense.

When comparing an experimental group with a control group, commonly the EG is better regarding almost every parameter measured than the CG. In this way I recommend to reconsider the idea of including another multicomponent-based training experimental group in the comparison (two experimental groups and a control group). In this way you could probe if your agility protocol is better than the traditional way of training nowadays, having a control group as a base reference.

Reply: We appreciate this note. As it was mentioned before, this is the protocol of an ongoint study. We thoroughly discussed this matter before the start of the study. Unfortunatelly, we did not have the personal and financial resources to integrate a second active control group due to relatively large sample sizes that were computed by sample size estimation. Nevertheless, we came to the conclusion that the comparison of our exercise intervention compared to a control group that does not receive any treatment, still adds relevant knowledge. The agility training framework that we propse, incluces a high variety of training aspects (strength, balance, endurance, cognition, agility as changes in velocity and direction) within only two hours of weekly training. Showing that this time investment is enough to account for improvements in all performance measures would be highly relevant and could have valuable impact on future training design for older adults. To our knowledge, no study has investigated a comparable training approach over a long-term period with the variety of measurements that we include. We therefore call it a “proof-of-concept training study” (line 94).

The difference of your multicomponent training and the other multicomponent training should be well defined.

Reply: Thank you for this comment. How we mark ourselves off from other multicomponent training is described in the introduction in line 68 et seqq.. We added “multidimensional, functional and time-efficient agility-training framework for older adults, that has not been evaluated in a randomized controlled trial to date” (line 69-70); “…it seems promision that agility training can…” (line 96); “To our knowledge, no randomized controlled long-term intervention has applied the agility framework yet.”(line 100-101)

If this is a protocol for RCT, the primary outcome should be only one variable. Please, revise.

Reply: We essentially agree with the point you make. Nevertheless, we think that for the multimodal intervention that addresses a huge variety of relevant parameters, choosing several parameters as primary outcomes is reasonable. Referring to the training, those primary outcomes comprise several domains, like strength, balance, gait and mobility. Those were also applied for sample size estimation.

I recommend to follow the SPIRIT checklist for protocols and to use the TIDIER checklist to describe the intervention.

Reply: Thanks for this note. We checked the SPIRIT checklist and are confident to have included all important aspects. Again, guided by another protocol of an ongoing study that was recently publised in this journal (Fischer et al. 2019), the order in which aspects are addressed migh differ sometimes. Aspects added were: modifications to the intervention “A participant does not continue the allocated intervention if he or she requests it or suffers pain or an injury.”(line 404-406); Implementation “MM is responsible for the organization of the study (recruitment, group allocation, data collection, training, quality control)” (line 499); Data management “Participant files are saved under non-peronalized identification numbers. A file for uncoding numbers is stored separately. Data entry is always checked by two researchers independently.” (line 181-182).

We also checked the TIDIER checklist and are confident to say that we included all relevant points either in the introduction, the experimental design description or the description of the intervention.

Apart from that, the study was registered in the german study register (drks) and the thus, the manuscript of this protokol also followed contents, given by the drks.

Reviewer 4 Report

In my opinion it is very good research paper about agility-training, which aim is to promote neuromuscular, cognitive, cardiovascular and psychosocial function in  elderly.

In general, this work meets the scope of International Journal of Environmental Research and Public Health and is of interest to the readership. The manuscript is quite well written.

Eighty-five participants were enrolled in this ongoing trial. Seventy-nine participants completed baseline testing and were block randomized to the agility-training group or the inactive control group. All participants undergo pre- and post-testing with interim assessment after six months. It was a one year randomized-controlled trial. The trial was registrated in German Clinical Trials Register.

Introduction briefly summarizes recent research related to the topic and gives a clear idea of the novelty. Materials and methods are understandable and typical for level Ib randomized controlled trial with low risk of bias. Sampling was sufficient to achieve goals set. Ethical standards were maintained.

 The Authors used the appropriate statistical methods.

The results are clearly presented and explained.  In the discussion the Authors presented reliable advantages and limitations of their study.  

The conclusions are consistent with presented evidence and arguments.

To sum up, in my opinion this manuscript may be published in this form. 

Author Response

Dear Editor, dear Reviewers,

Thank you very much for your positive feedback and your constructive comments. We appreciate your time and effort to improve the manuscript. In the following, we address all issues raised using a point-to-point reply. The point-to-point answers are in Italics, referring to the lines, where the changes have been made. We now feel that the manuscript notably improved and hope that the paper can be accepted for publication.

I hereby reconfirm the author list and the corresponding affiliations.

Thank you very much

Mareike Morat on behalf of all co-authors

Reviewer 4

In my opinion it is very good research paper about agility-training, which aim is to promote neuromuscular, cognitive, cardiovascular and psychosocial function in elderly.

In general, this work meets the scope of International Journal of Environmental Research and Public Health and is of interest to the readership. The manuscript is quite well written. Eighty-five participants were enrolled in this ongoing trial. Seventy-nine participants completed baseline testing and were block randomized to the agility-training group or the inactive control group. All participants undergo pre- and post-testing with interim assessment after six months. It was a one year randomized-controlled trial. The trial was registrated in German Clinical Trials Register. Introduction briefly summarizes recent research related to the topic and gives a clear idea of the novelty. Materials and methods are understandable and typical for level Ib randomized controlled trial with low risk of bias. Sampling was sufficient to achieve goals set. Ethical standards were maintained. The results are clearly presented and explained. In the discussion the Authors presented reliable advantages and limitations of their study. The conclusions are consistent with presented evidence and arguments.To sum up, in my opinion this manuscript may be published in this form.The Authors used the appropriate statistical methods.

Reply: Thanks a lot, we really appreciatie your feedback. It just needs to be mentioned, that this is a manuscript of a study protocol of an ongoint trail. Thus, no results are yet presented. Only expected key results are mentioned.

Round 2

Reviewer 3 Report

I think that the autor have made a good work.

It can be published in the present form.